# Anti-inflammatory and Neuroprotective Effects of Fungal Immunomodulatory Protein Involving Microglial Inhibition

**DOI:** 10.3390/ijms19113678

**Published:** 2018-11-21

**Authors:** Wen-Ying Chen, Cheng-Yi Chang, Jian-Ri Li, Jiaan-Der Wang, Chih-Cheng Wu, Yu-Hsiang Kuan, Su-Lan Liao, Wen-Yi Wang, Chun-Jung Chen

**Affiliations:** 1Department of Veterinary Medicine, National Chung Hsing University, Taichung 402, Taiwan; wychen@dragon.nchu.edu.tw; 2Department of Surgery, Feng Yuan Hospital, Taichung 402, Taiwan; c.y.chang.ns@gmail.com; 3Division of Urology, Taichung Veterans General Hospital, Taichung 407, Taiwan; fisherfishli@yahoo.com.tw; 4Department of Pediatrics & Child Health Care, Taichung Veterans General Hospital, Taichung 407, Taiwan; wangjiaander@gmail.com; 5Department of Anesthesiology, Taichung Veterans General Hospital, Taichung 407, Taiwan; chihcheng.wu@gmail.com; 6Department of Financial Engineering, Providence University, Taichung 433, Taiwan; 7Department of Data Science and Big Data Analytics, Providence University, Taichung 433, Taiwan; 8Department of Pharmacology, Chung Shan Medical University, Taichung 402, Taiwan; kuanyh001@gmail.com; 9Department of Medical Research, Taichung Veterans General Hospital, Taichung 407, Taiwan; slliao@vghtc.gov.tw; 10Department of Nursing, Hung Kuang University, Taichung 433, Taiwan; walice@sunrise.hk.edu.tw; 11Department of Medical Laboratory Science and Biotechnology, China Medical University, Taichung 404, Taiwan

**Keywords:** immunomodulatory protein, microglia, neuroinflammation, polarization

## Abstract

Microglia polarization of classical activation state is crucial to the induction of neuroinflammation, and has been implicated in the pathogenesis of numerous neurodegenerative diseases. Fungal immunomodulatory proteins are emerging health-promoting natural substances with multiple pharmacological activities, including immunomodulation. Herein, we investigated the anti-inflammatory and neuroprotective potential of fungal immunomodulatory protein extracted from *Ganoderma microsporum* (GMI) in an in vitro rodent model of primary cultures. Using primary neuron/glia cultures consisting of neurons, astrocytes, and microglia, a GMI showed an alleviating effect on lipopolysaccharide (LPS)/interferon-γ (IFN-γ)-induced inflammatory mediator production and neuronal cell death. The events of neuroprotection caused by GMI were accompanied by the suppression of Nitric Oxide (NO), Tumor Necrosis Factor-α (TNF-α), Interleukin-1β (IL-1β), and Prostaglandin E2 (PGE2) production, along with the inhibition of microglia activation. Mechanistic studies showed that the suppression of microglia pro-inflammatory polarization by GMI was accompanied by the resolution of oxidative stress, the preservation of protein tyrosine phosphatase and serine/threonine phosphatase activity, and the reduction of NF-κB, AP-1, cyclic AMP response element-binding protein (CREB), along with signal transducers and activators of transcription (Stat1) transcriptional activities and associated upstream activators. These findings suggest that GMI may have considerable potential towards the treatment of neuroinflammation-mediated neurodegenerative diseases.

## 1. Introduction

Neuronal dysfunction and/or death are common events contributing to both acute and chronic neurodegenerative diseases, including Parkinson’s disease, Alzheimer’s disease, and stroke. Among the etiology and pathogenesis responsible for neuronal degeneration, neuroinflammation is an emerging mechanism for the initiation and progression of degeneration. Activation of resident glial cells, particularly the microglia, along with the recruitment of peripheral leukocytes, and the overproduction of inflammatory mediators within the nervous system, form the basis of neuroinflammation. These neuroinflammatory responses are present in the vicinity of degenerating neurons, cerebrospinal fluids, and even systemic blood circulation [1,2,3]. The line of evidence indicates that peripheral inhibition of inflammatory responses also brings about benefits in cases of neurodegenerative diseases [4,5]. The aforementioned phenomena underscore the feasibility of anti-inflammatory strategies in the prevention and therapeutic treatment against neurodegenerative diseases.

On the basis of cell levels, neurons, astrocytes, microglia, oligodendrocytes, pericytes, and other minor cell types orchestrate the structural and functional units of the nervous system. Among those component cells, microglia in particular show remarkable immune competency and are intimately associated with neuronal function and fate. Adequate microglial activation modulates both synaptic function and plasticity, while maintaining tissue homeostasis. Once overactivated, microglia release a wide range of inflammatory and bioactive molecules which impose detrimental impacts on neurons [1,2,3,6,7]. Therefore, microglia represent important targets for therapeutic intervention, with the aim being to combat neurodegenerative diseases.

Beyond being a basic food and nutrition source, mushrooms are an important source of bioactive compounds. Amongst them, *Ganoderma lucidum* has been widely used as both an alternative medicinal remedy and dietary supplement in order to promote health and longevity. *Ganoderma lucidum*-derived spores, extracts, triterpenoids, and polysaccharides have a wide range of biological activities, including neuroprotection. Those extracts and ingredients alleviate microglial activation both in vitro and in vivo, as well as protect the brain against lipopolysaccharide-, streptozotocin-, Aβ-, 1-methyl-4-phenyl-1,2,3,6-tetrahydropyridine (MPTP)-, and cerebral ischemia-induced neuronal degeneration [8,9,10,11,12]. Additionally, Fungal Immunomodulatory Proteins (FIP) with a molecular weight of approximately 12 kDa, termed LZ-8, represent alternative components isolated from this mushroom with pleiotropic bioactivities, including immunomodulation and antitumor [13,14]. FIPs have also been isolated and purified from *Ganoderma microsporum* (GMI), *Ganoderma tsugae* (FIP-gts), and *Flammulina velutipes* (FIP-*fve*). The immunomodulatory effects of FIPs have been implicated in viral inflammation, allergies, pulmonary inflammation, and asthma [15,16,17,18,19,20,21,22]. Therefore, FIPs may possess the potential to prevent and/or treat inflammation-associated diseases, including neurodegenerative diseases.

Although *Ganoderma lucidum* has been implicated in anti-neuroinflammation and neuroprotection [8,9,10,11,12], the potential role and involvement of FIPs in neuroprotection are yet to be elicited. GMI, an immunomodulatory protein obtained from *Ganoderma microsporum*, contains 111 amino acids, with the amino acid sequence highly homologous with those of the other FIPs, including LZ-8. Evidence indicates that purified and recombinant GMI display remarkable anti-inflammatory and antitumor effects [21,23]. To extend the scope of FIP biological activities and get insight into their translational implication, the aims of this study were to investigate whether GMI exerted anti-inflammatory and neuroprotective effects via rodent primary cultures, and if so, to further determine the regulatory characteristics of the beneficial molecular responses.

## 2. Results

### 2.1. GMI Alleviated Neuronal Death

Neuron/glia cultures consisted of neurons, astrocytes, and microglia. LPS/IFN-γ treatment caused degeneration of neurite processes and decreased numbers of visible neurons, as evidenced by immunocytochemical staining for neuron-associated cytoskeletal MAP-2 protein. The disruption of neuronal morphology and integrity along with decreased number of viable neurons were alleviated by GMI (Figure 1A). Upon LPS/IFN-γ exposure, microglia increased cell number and changed into a rounded, darkly stained cell morphology. The alternations in microglia were alleviated by GMI (Figure 2A). However, the morphological integrity of astrocytes within neuron/glia cultures was not compromised by LPS/IFN-γ (Figure 2B). Neuronal damage was further examined by measuring MAP-2 protein content and LDH efflux. LPS/IFN-γ treatment caused a reduction of MAP-2 protein content (Figure 1B) and an increase of LDH efflux (Figure 1C). Protein tyrosine kinase inhibitor genistein is a neuroprotectant [24]. GMI alleviated MAP-2 protein loss (Figure 1B) and LDH efflux (Figure 1C) as that did by genistein. These findings indicate that LPS/IFN-γ induced selective neuronal death within neuron/glia cultures and GMI showed neuroprotective effects.

### 2.2. GMI Alleviated Inflammatory Cytokine and Neurotoxic Mediator Production

Among the neuroprotective mechanisms, anti-inflammation is crucial to the actions of genistein [24]. LPS/IFN-γ treatment caused a robust production of pro-inflammatory mediators such as NO, TNF-α, IL-1β, and PGE2 in neuron/glia cultures. Genistein attenuated LPS/IFN-γ-induced pro-inflammatory mediator production (Figure 3). GMI alleviated the production of NO, TNF-α, IL-1β, and PGE2, as well (Figure 3). Microglia are considered to be a determinant cell type responsible for neuroinflammation, more so than neurons and astrocytes [6,7,25,26,27]. Mixed glia were composed of astrocytes and microglia. As with neuron/glia cultures, LPS/IFN-γ treatment induced pro-inflammatory mediator production in mixed glia (Figure 4) and microglia (Figure 5) cultures, while their elevated production was alleviated by genistein and GMI. In comparison, astrocytes were relatively inert to the actions of LPS/IFN-γ and GMI (Figure 6). Results of our findings show an anti-inflammatory effect of GMI against LPS/IFN-γ-induced neuroinflammation, and microglia may be the target of action.

### 2.3. GMI Alleviated Microglial Activation

The pro-inflammatory phenotypes of microglia are characterized by several upstream activators and downstream effectors, including IRF5, IRF8, P2X4R, P2X7R, P2Y12R, iNOS, COX-2, and CD68 [28,29,30]. In neuron/glia cultures, LPS/IFN-γ elevated the protein levels of IRF5, IRF8, P2X4R, P2X7R, P2Y12R, iNOS, COX-2, and CD68. The levels of microglia reactive phenotype-associated activators and effectors were alleviated by GMI (Figure 7). These findings imply that alleviation of pro-inflammatory phenotype of microglia may be attributed to the anti-inflammatory effect of GMI.

### 2.4. GMI Alleviated Inflammatory Transcription Factors

NF-κB, AP-1, CREB, and Stat are latent transcription factors critical to the acquisition of pro-inflammatory phenotype of microglia, and the transcriptional activation of inflammatory mediator expression [6,7,25,26,27,28,29,30]. Their DNA binding activities and corresponding action axes for activation in neuron/glia cultures were assessed by EMSA and Western blotting. An elevated protein phosphorylation in IKK-α/β, and p65 (Figure 8A), along with the DNA binding activity of NF-κB (Figure 8B) was noted in LPS/IFN-γ-stimulated cultures. Likewise, LPS/IFN-γ treatment caused an increase of protein phosphorylation in c-Jun, CREB, and Stat1 (Figure 8A), protein expression in c-Fos and IRF1 (Figure 8A), and DNA binding activity in AP-1, CREB, and Stat1 (Figure 8B). GMI had an alleviating effect on the activation of NF-κB, AP-1, CREB, and Stat1 signaling (Figure 8). These findings imply that the anti-inflammatory effects of GMI may be attributed to its inhibition on axes of NF-κB, AP-1, CREB, and Stat1 signaling.

### 2.5. GMI Alleviated Inflammatory Intracellular Signaling

The activation of NF-κB, AP-1, CREB, and Stat1 axes is multifactorial, and includes adaptors and intracellular signaling molecules MyD88, TAK1, TBK1, ERK, JNK, p38, Akt, Jak1, Tyk2, and Src as well as free radicals [7,24,27,31,32]. Treatment of neuron/glia cultures with LPS/IFN-γ increased protein expression in MyD88 and protein phosphorylation in TAK1, TBK1, ERK, JNK, p38, Akt, Jak1, Tyk2, and Src (Figure 9A), as well as free radical production (Figure 9B). In the presence of GMI, there was a decline of protein expression, protein phosphorylation (Figure 9A), and free radical production (Figure 9B). To investigate the involvement of intracellular signaling molecules in neuronal cell death and neuroinflammation, corresponding pharmacological inhibitors were applied. NF-κB inhibitor [3-chloro-4-nitro-*N*-(5-nitro-2-thiazolyl)-benzamide; SM-7368], ERK inhibitor (U0126), JNK inhibitor (SP600125), p38 inhibitor (SB203580), Akt inhibitor (LY294002), Jak inhibitor (AG490), Src inhibitor (PP2), COX-2 inhibitor (NS398), and antioxidant (*N*-acetyl-cystein, NAC) all showed alleviating effects on LPS/IFN-γ-induced NO, TNF-α, IL-1β, and PGE2 production (Figure 10), as well as neuronal cell death (Figure 11). This implies that the anti-inflammatory and neuroprotective effects of GMI may involve the inhibition of ERK, JNK, p38, Akt, Jak, Src, COX-2, NF-κB, and oxidative stress.

### 2.6. GMI Alleviated Protein Phosphatase Inactivation

Aforementioned protein phosphorylation events in intracellular signaling molecules are governed by the action balance between kinases and phosphatases. Through enzymatic assays, we found that LPS/IFN-γ treatment caused a decline in protein tyrosine phosphatase activity (Figure 12A) and serine/threonine phosphatase activity (Figure 12B) with the inactivation alleviated by GMI (Figure 12). These findings suggest that the downregulation of protein phosphorylation in intracellular signaling molecules by GMI may be partially attributed to the preservation of protein phosphatase activity.

## 3. Discussion

The fungi *Ganoderma lucidum* is a traditional Oriental medicine well-known for its health promotion and disease prevention and/or treatment. For some time now, triterpenoids, polysaccharides, and other small molecules are believed to be the attributing factors to its described biological activities [8,9,10,11,12]. Increasing evidence indicates that protein components are also being seen as active molecules of *Ganoderma lucidum* and extend to other edible mushrooms, making them responsible for their beneficial effects [13,14,15,16,17,18,19,20,21,22]. Herein, we found that the GMI immunomodulatory protein of *Ganoderma microsporum* displayed both anti-inflammatory and neuroprotective effects in in vitro cell models. Using rodent primary neuron/glia cultures consisting of neurons, astrocytes, and microglia, GMI proved to have an alleviating effect on LPS/IFN-γ-induced inflammatory mediator production and neuronal cell death. Biochemical studies revealed that bystander inflammatory damage caused by microglia was partly the action target of GMI for the neuroprotection. The actions of several transcription factors and intracellular signaling molecules, including NF-κB, AP-1, CREB, Stat1, IRF1, IRF5, IRF8, P2X4R, P2X7R, P2Y12R, TAK1, TBK1, ERK, JNK, p38, Akt, Jak1, Tyk2, Src, along with free radicals favoring pro-inflammatory phenotype of microglia were reversed by GMI. Additionally, GMI alleviated LPS/IFN-γ-induced inactivation of protein tyrosine phosphatase and serine/threonine phosphatase activities. Therefore, the pharmacological mechanisms underlying GMI’s neuroprotection are multifactorial, at the least involving the reversal of pro-inflammatory phenotype of microglia.

The action of neuroinflammation acts likes a double-edged sword. Immune surveillance, pathogen clearance, and tissue remodeling all work together in concert to keep the homeostasis of the nervous system. However, uncontrolled neuroinflammation has been implicated in the initiation and progression of degeneration through a bystander mechanism [1,2,3]. Independent of central or peripheral origin, overproduction of inflammatory mediators such as NO, TNF-α, IL-1β, and PGE2 shows apparent neuron-killing activity. No matter what kind of strategies are implemented, the suppression of NO, TNF-α, IL-1β, along with PGE2 overproduction and accumulation, protects neurons from neuroinflammation-mediated degeneration [4,5,24,26,33]. Accumulated studies further highlight the critical role and importance of microglia in the production of NO, TNF-α, IL-1β, and PGE2 and underscore microglia as the target of intervention for the resolution of neuroinflammation [6,7,25,27,33]. The event of neuroprotection caused by GMI was accompanied by the suppression of NO, TNF-α, IL-1β, and PGE2 production, along with inhibition of microglia activation. Results of both biochemical and pharmacological studies showed a likely microglia origin for the production and neurotoxic consequences of NO, TNF-α, IL-1β, and PGE2. There were have been studies showing the anti-inflammatory and neuroprotective effects of *Ganoderma lucidum*-derived triterpenoids, polysaccharides, and other small molecules by acting on microglia [8,9,10,11,12]. In this study, we provided further evidence demonstrating that the immunomodulatory protein GMI of *Ganoderma microsporum* also acted on microglia to show neuroprotection. Therefore, microglia may be a key target for mushroom products to exert biological activities on the central nervous system.

Microglia are typically classified into the classical activation state and alternative activation state, characterized by pro-inflammatory and anti-inflammatory phenotypes, respectively. Classical activation polarized microglia are a prominent source of pro-inflammatory cytokines such as TNF-α, IL-1β, IL-18, IL-6, and IFN-γ, as well as neurotoxic mediators such as NO, PGE2, and free radicals. Microglia in the alternative activation state, in contrast, have the ability to release Transforming Growth Factor (TGF-β), IL-4, and IL-10, with the consequences of suppressing inflammation and restoring homeostasis. Moreover, IRF5, IRF8, P2X4R, P2X7R, and P2Y12R promote microglia polarization towards the classical activation state. By contrast, CD206 (a mannose receptor) and arginase 1 polarize microglia towards the alternative activation state [28,29,30,34,35]. Alterations in microglia polarization favoring the classical activation state are dominant in numerous neurodegenerative diseases. A reversal balance by inhibiting pro-inflammatory phenotype resolves neuroinflammation and ameliorates neuronal degeneration [1,2,3,34,35]. Upon LPS/IFN-γ stimulation, microglia were polarized towards the classical activation state by increasing the expression of polarization transcription factors and downstream inflammatory cytokines and neurotoxic mediators. The effectors and biomarkers of microglia classical activation state were alleviated by GMI. This means that the reversal of microglia pro-inflammatory polarization and a switch towards the alternative activation state may be an action mechanism of GMI to ameliorate neuroinflammation. Currently, the phenotypes of the alternative activation state of microglia are not addressed in detail.

LPS/IFN-γ is commonly used to induce the microglia classical activation state in vitro. Upon exposure to LPS, the engagement with the transmembrane Toll-like Receptor 4 (TLR4) turns on intracellular phosphorylatory cascades to spread diverse signals, and eventually impact on the nuclear influx activity, DNA binding activity, and transcriptional activity of transcription factors, including NF-κB, AP-1, CREB, and Stat1. The linking adaptors and signaling molecules include MyD88, TAK1, TBK1, mitogen-activated protein kinases (ERK, JNK, and p38), IKK-α/β, p65, c-Jun, c-Fos, CREB, Src, as well as Akt. Likewise, the binding of IFN-γ with receptors activates Stat1 to drive IRF1 expression, by which Stat1 and IRF1 work in concert to boost another wave of transcriptional program. The activation of the Stat family latent transcriptional factors is promoted by upstream stimulatory tyrosine kinases such as Jak and Src family kinases, and inhibited by Src Homology 2 domain containing Phosphatase (SHP), Suppressor of Cytokine Signaling (SOCS), and Protein Inhibitors of Activated Stats (PIAS) [6,7,10,25,31]. GMI alleviated LPS/IFN-γ-activated downstream adaptors, signaling molecules, and transcription factors. Pharmacological inhibitors corresponding to several steps of LPS/IFN-γ intracellular actions caused a reduction in the production of inflammatory cytokines and neurotoxic mediators, as well as in neuronal cell death. Other than microglia, GMI downregulated TNF-α-induced NF-κB activation and gene expression in A549 epithelial cells [21]. These phenomena highlight the role of GMI in suppressing intracellular inflammatory signaling.

Free radicals are commonly multiplied during the course of neuroinflammation. Apart from being a primary neurotoxic mediator, free radicals indirectly cause neuronal cell death through their action on redox-sensitive intracellular signaling molecules and transcription factors to augment neuroinflammation. Moreover, oxidative stress represents an alternative mechanism to cause inactivation of protein phosphatase [6,7]. GMI-mediated anti-inflammation and neuroprotection were paralleled with the resolution of oxidative stress, reduction of signaling molecule phosphorylation, and preservation of protein tyrosine phosphatase and serine/threonine phosphatase activities. Our data showed that one of the pharmacological activities of GMI is its antioxidant effect.

Biological products are emerging as therapeutic drugs in human health care. In traditional Oriental medicine, plants are a good source of small molecular compounds. The identification and translational application of LZ-8 immunomodulatory proteins of *Ganoderma lucidum* offer another view on the health-promoting effects of natural substances [13,14]. Daily oral administration of Fungal Immunomodulatory Proteins protected rodents from inflammatory diseases [15,16,17,18,19,20]. Evidence also indicates that Fungal Immunomodulatory Proteins are resistant to digestive enzymes in stimulated gastric fluid and stimulated intestinal fluid [36]. Although a blood-brain barrier restricts the peripheral materials from targeting the central nervous system, its integrity is compromised in neurodegenerative diseases. During the pathological progression of stroke, for example, the compromised blood-brain barrier allows Evans blue-albumin complexes to enter the brain [1]. Otherwise, the infiltration of peripheral immune cells into the central nervous system exacerbates the pathogenic progression of neurodegenerative diseases [4,5]. Findings from these relevant studies suggest that GMI may have anti-inflammatory and neuroprotective effects to help combat neuroinflammation-mediated neurodegenerative diseases through their action upon either peripheral immune cells or central microglia. However, the in vivo and central effects of GMI warrant our continuous investigation.

Our results clearly demonstrate that the anti-inflammatory and neuroprotective effects of GMI are mediated through the inhibition of pro-inflammatory phenotype of microglia in the model of LPS/IFN-γ-stimulated neuron/glia cultures. The suppression of microglia pro-inflammatory polarization and a switch towards the alternative activation state by GMI was accompanied by the resolution of oxidative stress, preservation of protein tyrosine phosphatase and serine/threonine phosphatase activity, and reduction of NF-κB, AP-1, CREB, and Stat1 transcriptional activities and associated upstream activators. These findings suggest that GMI may have considerable potential for the treatment of neuroinflammation-mediated neurodegenerative diseases.

## 4. Materials and Methods

### 4.1. Cell Cultures

All experimental protocols for animal study were reviewed and approved by the Animal Experimental Committee of Taichung Veterans General Hospital (La-93146, 15 Dec 2004) and subsequently conducted in strict accordance with the guidelines. Neuron/glia, mixed glia, and microglia cultures were prepared from cerebral cortices of postnatal day 1 Sprague-Dawley rats in accordance with previously reported methods [26,27,33]. After physical trituration and enzymatic digestion, the dissociated cells from the dissected cortices were plated onto poly-d-lysine-coated dishes. For the preparation of neuron/glia, the cells were maintained in a minimum essential medium, supplemented with 10% Fetal Bovine Serum (FBS) and 10% horse serum for 10–12 days in vitro. Mixed glia were maintained in Dulbecco’s Modified Eagle Medium (DMEM)/F12 containing 10% FBS for 14 days in vitro. Microglia were isolated from confluent mixed glia through shaking at a speed of 200 rpm for 24 h, and maintained in DMEM/F12 containing 10% FBS. The retained astrocytes were replated and maintained in DMEM/F12 containing 10% FBS. Neuron/glia consisted of 30–40% neurons, 40–50% astrocytes, and 10–15% microglia. Mixed glia were composed of ~85% astrocytes and ~15% microglia. The purity of microglia and astrocytes was more than 95%. Cell composition was identified and estimated by immunocytochemical staining with antibodies recognizing neuron (Microtubule-associated Protein 2 (MAP-2), Transduction Laboratories, Lexington, KY, USA), astrocyte (Glial Fibrillary Acidic Protein, GFAP, Santa Cruz Biotechnology, Santa Cruz, CA, USA), and microglia (CD68, Santa Cruz Biotechnology, Santa Cruz, CA, USA), respectively.

### 4.2. GMI Proteins

Recombinant GMI proteins were manufactured by Mycomagic Biotechnology Co., Ltd. (Taipei, Taiwan). The detailed procedures for GMI cloning, expression, purification, and verification were described as in previously reported methods [21,23].

### 4.3. Cytotoxicity Assessment

A quantitative measurement of Lactate Dehydrogenase (LDH) released into the culture media by the Pierce^™^ LDH Cytotoxicity Assay Kit (Thermo Fisher Scientific, Waltham, MA, USA) was used to assess cell damage in accordance with the manufacturer’s instructions. The cytotoxicity index was indicated by the ratio of released LDH and total LDH.

### 4.4. Immunocytochemical Staining

For the conduction of immunocytochemical staining, neuron/glia (24-well plates) were washed with Phosphate-buffered Saline (PBS), fixed with 4% paraformaldehyde, and permeabilized with 0.1% Triton X-100, in accordance with our previously reported methods [26,27,33]. After blocking with 5% nonfat milk, the cells were incubated sequentially with antibodies against MAP-2 (Transduction Laboratories, Lexington, KY, USA), GFAP (Santa Cruz Biotechnology, Santa Cruz, CA, USA), or CD68 (Santa Cruz Biotechnology, Santa Cruz, CA, USA), while horseradish peroxidase conjugated the secondary antibody. The immunoreactive signals were developed with 3, 3′-diaminobenzidine (DAB) and observed using a light microscope. For the quantification of immuno-positive cell number, the total number of reactive cells was counted from randomly selected four fields in a well of a 24-well plate. Four replicates were conducted for each experiment.

### 4.5. Enzyme-linked Immunosorbent Assay (ELISA)

Neuron/glia, mixed glia, and microglia were seeded onto 24-well plates. For the conduction of ELISA, supernatants were collected and subjected to measurement using commercial ELISA kits, including Tumor Necrosis Factor-α (TNF-α), Interleukin-1β (IL-1β), and Prostaglandin E2 (PGE2), according to the manufacturer’s instructions (R&D Systems, Minneapolis, MN, USA).

### 4.6. Nitric Oxide (NO) Determination

For NO (nitrite/nitrate) determination, neuron/glia, mixed glia, and microglia were seeded onto 24-well plates. The supernatants were collected and subjected to measurement using a Griess Reagent Kit, according to the manufacturer’s instructions (Thermo Fisher Scientific, Waltham, MA, USA).

### 4.7. Free Radical Determination

The levels of intracellular free radicals were assayed by measuring intracellular oxidation of dichlorofluorescein, as in our previously described methods [7]. Neuron/glia were seeded onto 24-well plates. Cultures were loaded with 10 μM cell-permeant 2′, 7′-Dichlorodihydrofluorescein Diacetate (H2DCFDA) at 37 °C for 1 h. The fluorescence signal of oxidized 2′, 7′-dichlorodihydrofluorescein was measured using a fluorometer (E_x_ 485 nm and E_m_ 510 nm).

### 4.8. Western Blot Analysis

To conduct Western blot analysis, neuron/glia were seeded onto 6-well plates. Cultured cells were homogenized with a Laemmli SDS sample buffer, and the resultant cell lysates were subjected to SDS-PAGE separation and PVDF electrophoretical transfer [7]. The membranes were incubated in sequence with primary antibodies and horseradish peroxidase-labeled IgG. The immunoreactive signals in the membranes were visualized by enhanced chemiluminescence Western blotting reagents and quantitated using a computer image analysis system (IS1000; Alpha Innotech Corporation, San Leandro, CA, USA). Quantitative data were depicted under blots and the content in untreated control was defined as 1.0. Antibodies recognizing specific protein were listed as follows: MyD88, phosphorylated transforming growth factor β-activated kinase-1 (TAK1), TAK1, phosphorylated TANK-binding kinase (TBK1), TBK1, phosphorylated IκB kinase (IKK-α/β), IKK-α/β, phosphorylated p65, p65, phosphorylated extracellular signal-regulated kinase (ERK), ERK, phosphorylated c-Jun N-terminal kinase (JNK), JNK, phosphorylated p38, p38, phosphorylated Akt, Akt, phosphorylated Src, Src, phosphorylated Janus kinase 1 (Jak1), Jak1, phosphorylated Tyk2, Tyk2, phosphorylated signal transducers and activators of transcription 1 (Stat1), Stat1, Interferon Regulatory Factor 1 (IRF1), IRF5, IRF8, P2X purinoceptor 4 (P2X4R), P2X7R, P2Y12R, CD68, inducible Nitric Oxide Synthase (iNOS), Cyclooxygenase 2 (COX-2), phosphorylated c-Jun, c-Jun, c-Fos, Glyceraldehyde 3-phosphate Dehydrogenase (GAPDH) (Santa Cruz Biotechnology, Santa Cruz, CA), MAP-2 (Transduction Laboratories, Lexington, KY), phosphorylated cAMP Response Element Binding protein (CREB), and CREB (Epitomics, Burlingame, CA, USA).

### 4.9. Preparation of Nuclear Extracts and Electrophoretic Mobility Shift Assay (EMSA)

Neuron/glia were seeded onto 6-well plates. A commercially available nuclear extraction kit and EMSA kit were used to prepare nuclear extracts and conduct EMSA, respectively, according to the manufacturer’s instructions (Panomics, Fremont, CA, USA). As with our previous studies [6,7,33], nuclear extracts (5 μg) were reacted with NF-κB oligonucleotide (5′-AGTTGAGGGGACTTTCCCAGGC), AP-1 oligonucleotide (5′-CGCTTGATGAGTCAGCCGGAA), CREB oligonucleotide (5′-AGAGATTGCCTGACGTCAGAGAGCTAG), or Stat1 oligonucleotide (5′-ATCGTTCATTTCCCGTAAATCCCTA). The DNA/protein complexes in the membranes were visualized by enhanced chemiluminescence Western blotting reagents and quantitated using a computer image analysis system (IS1000; Alpha Innotech Corporation). Quantitative data were depicted under blots and the content in untreated control was defined as 1.0.

### 4.10. Statistical Analysis

All statistical results are presented as the mean ± standard deviation. A one-way analysis of variance was performed to evaluate experimental values between groups with a consequent Dunnett’s test or Tukey post-hoc test performed for the purpose of comparison. It was considered statistically significant when the *p* value was less than 0.05.

## Figures and Tables

**Figure 1 ijms-19-03678-f001:**
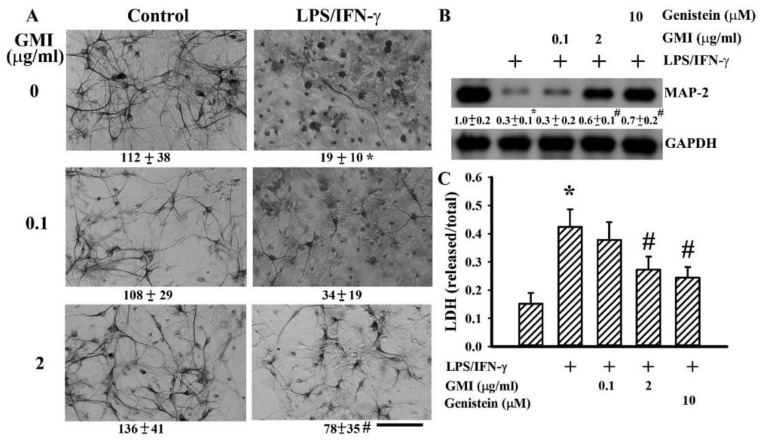
*Ganoderma microsporum* (GMI) alleviated neuronal cell death. (**A**) Neuron/glia cultures were pretreated with vehicle or various concentrations of GMI (0.1 and 2 μg/mL) for 30 min before being incubated with LPS (100 ng/mL)/IFN-γ (10 U/mL) for an additional 48 h. Neuronal viability was detected by immunocytochemical staining of MAP-2. Representative micrographs and quantitative numbers are shown. Bar, 60 μm. Neuron/glia cultures were pretreated with vehicle, various concentrations of GMI (0.1 and 2 μg/mL), or genistein (10 μM) for 30 min before being incubated with LPS (100 ng/mL)/IFN-γ (10 U/mL) for an additional 48 h. Total cellular proteins were extracted and subjected to Western blot analysis with indicated antibodies (**B**). One representative blot of four independent culture batches is shown and the fold of relative protein content is depicted under the blots. The protein content of untreated control was defined as 1.0. Cell damage was measured by LDH efflux assay (**C**). * *p* < 0.05 vs. untreated control and # *p* < 0.05 vs. LPS/IFN-γ control, *n* = 4. The + means the groups treated with LPS/IFN-γ and the rest groups without + indicate vehicle treatment.

**Figure 2 ijms-19-03678-f002:**
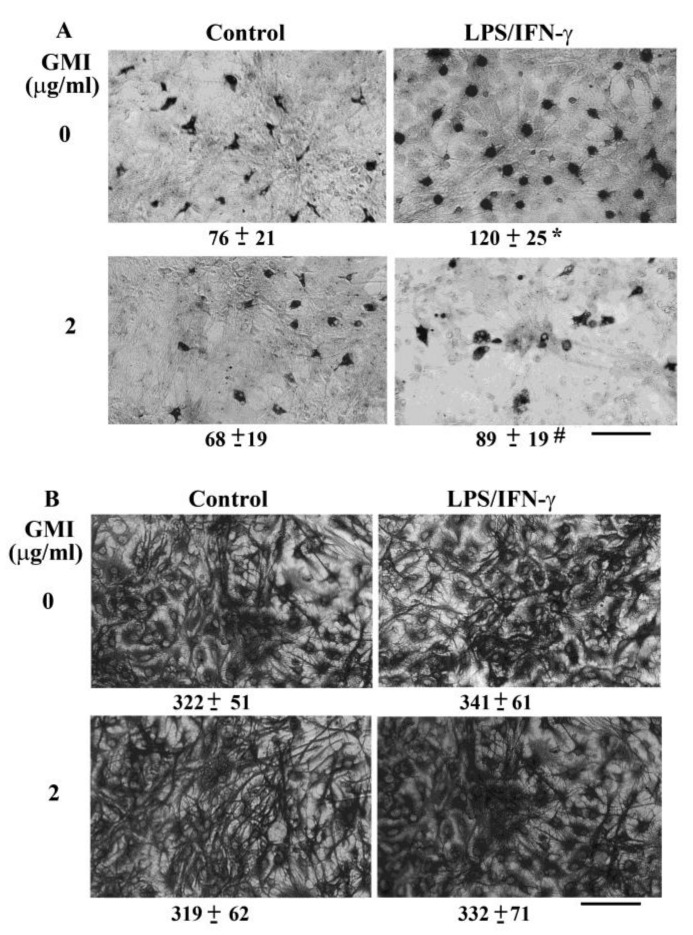
GMI alleviated microgliosis. Neuron/glia cultures were pretreated with vehicle or GMI (2 μg/mL) for 30 min before being incubated with LPS (100 ng/mL)/IFN-γ (10 U/mL) for an additional 48 h. Microglia (**A**) and astrocytes (**B**) were examined by immunocytochemical staining of CD68 and GFAP, respectively. Representative micrographs and quantitative numbers are shown. Bar, 60 μm. * *p* < 0.05 vs. untreated control and # *p* < 0.05 vs. LPS/IFN-γ control, *n* = 4.

**Figure 3 ijms-19-03678-f003:**
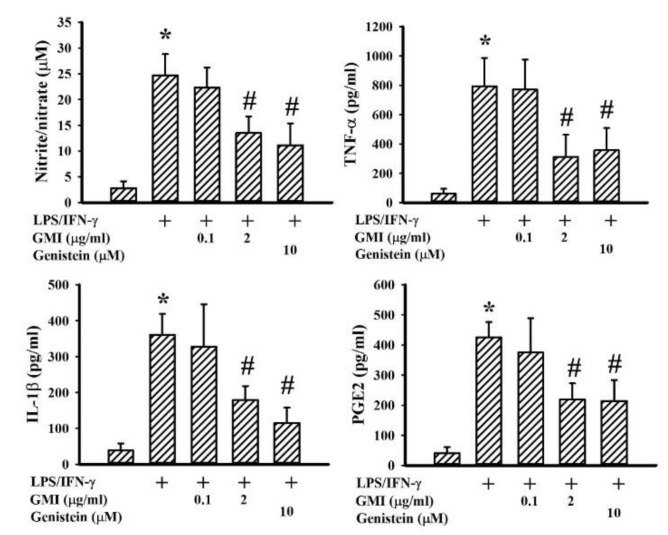
GMI alleviated inflammatory mediator production in neuron/glia cultures. Neuron/glia cultures were pretreated with vehicle, various concentrations of GMI (0.1 and 2 μg/mL), or genistein (10 μM) for 30 min before being incubated with LPS (100 ng/mL)/IFN-γ (10 U/mL) for an additional 24 h. Supernatants were collected and subjected to Griess reagent or ELISA for the measurement of NO, TNF-α, IL-1β, and PGE2. * *p* < 0.05 vs. untreated control and # *p* < 0.05 vs. LPS/IFN-γ control, *n* = 4. The + means the groups treated with LPS/IFN-γ and the rest groups without + indicate vehicle treatment.

**Figure 4 ijms-19-03678-f004:**
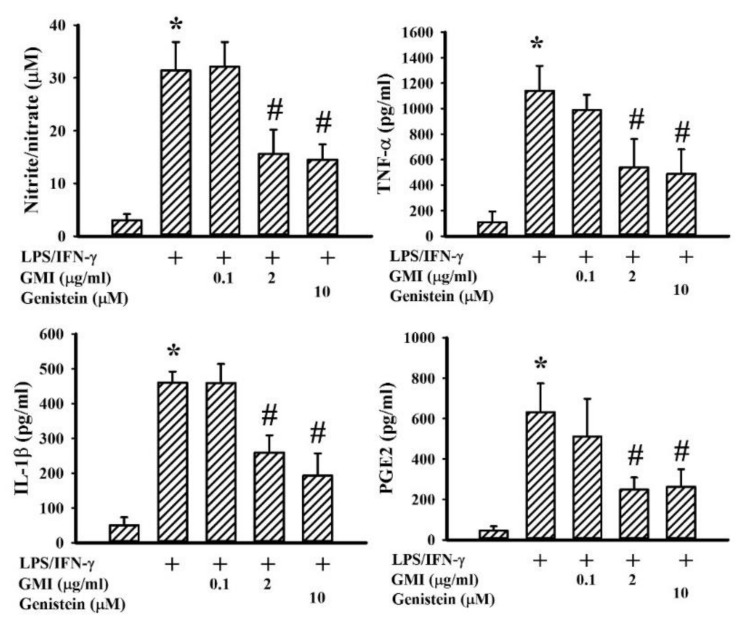
GMI alleviated inflammatory mediator production in mixed glia cultures. Mixed glia cultures were pretreated with vehicle, various concentrations of GMI (0.1 and 2 μg/mL), or genistein (10 μM) for 30 min before being incubated with LPS (100 ng/mL)/IFN-γ (10 U/mL) for an additional 24 h. Supernatants were collected and subjected to Griess reagent or ELISA for the measurement of Nitric Oxide (NO), TNF-α, IL-1β, and PGE2. * *p* < 0.05 vs. untreated control and # *p* < 0.05 vs. LPS/IFN-γ control, *n* = 4. The + means the groups treated with LPS/IFN-γ and the rest groups without + indicate vehicle treatment.

**Figure 5 ijms-19-03678-f005:**
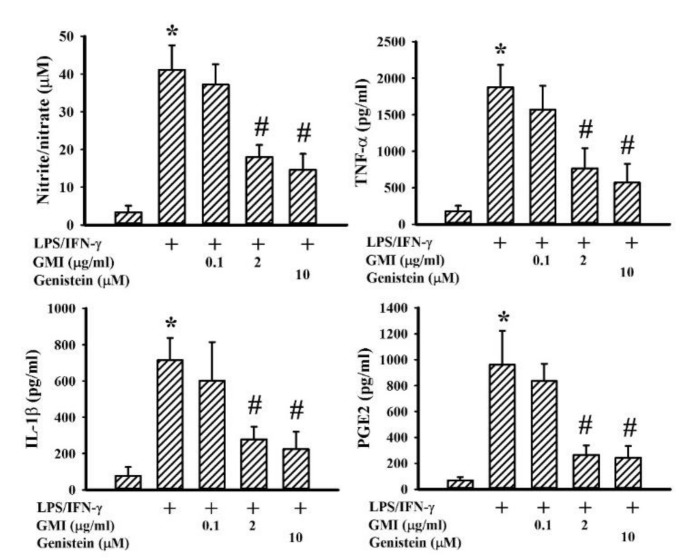
GMI alleviated inflammatory mediator production in microglia cultures. Microglia cultures were pretreated with vehicle, various concentrations of GMI (0.1 and 2 μg/mL), or genistein (10 μM) for 30 min. before being incubated with LPS (100 ng/mL)/IFN-γ (10 U/mL) for an additional 24 h. Supernatants were collected and subjected to Griess reagent or ELISA for the measurement of NO, TNF-α, IL-1β, and PGE2. * *p* < 0.05 vs. untreated control and # *p* < 0.05 vs. LPS/IFN-γ control, *n* = 4. The + means the groups treated with LPS/IFN-γ and the rest groups without + indicate vehicle treatment.

**Figure 6 ijms-19-03678-f006:**
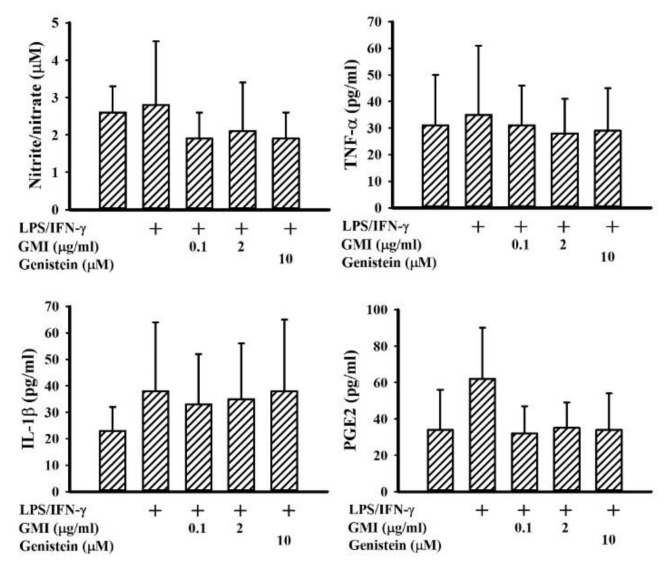
LPS/IFN-γ and GMI failed to alter inflammatory mediator production in astrocyte cultures. Astrocyte cultures were pretreated with vehicle, various concentrations of GMI (0.1 and 2 μg/mL), or genistein (10 μM) for 30 min. before being incubated with LPS (100 ng/mL)/IFN-γ (10 U/mL) for an additional 24 h. Supernatants were collected and subjected to Griess reagent or ELISA for the measurement of NO, TNF-α, IL-1β, and PGE2. *n* = 4. The + means the groups treated with LPS/IFN-γ and the rest groups without + indicate vehicle treatment.

**Figure 7 ijms-19-03678-f007:**
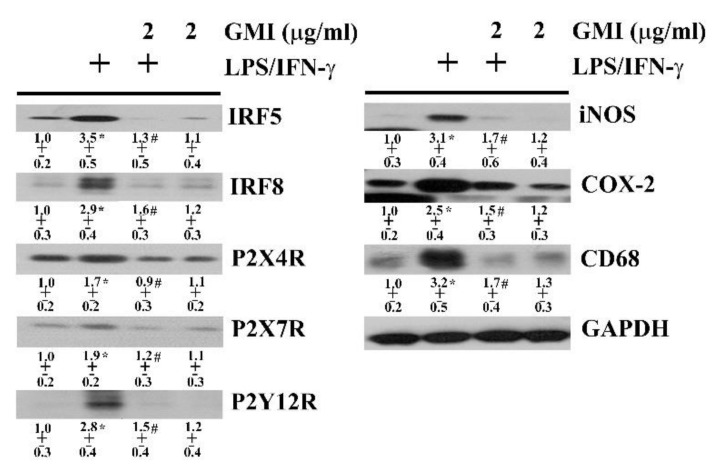
GMI alleviated microglial activation. Neuron/glia cultures were pretreated with vehicle or various concentrations of GMI (0.1 and 2 μg/mL) for 30 min before being incubated with LPS (100 ng/mL)/IFN-γ (10 U/mL) for an additional 24 h. Total cellular proteins were extracted and subjected to Western blot analysis with indicated antibodies. One representative blot of four independent culture batches is shown and the fold of relative protein content is depicted under the blots. The protein content of untreated control was defined as 1.0. The + means the groups treated with LPS/IFN-γ and the rest groups without + indicate vehicle treatment. * *p* < 0.05 vs. untreated control and # *p* < 0.05 vs. LPS/IFN-γ control.

**Figure 8 ijms-19-03678-f008:**
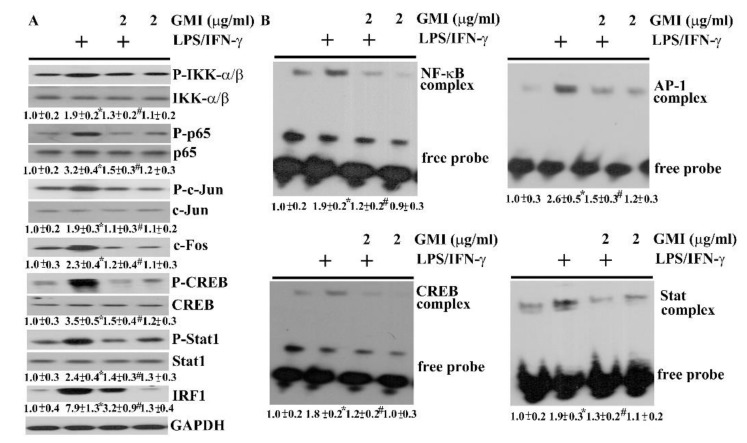
GMI alleviated transcription factor activation. Neuron/glia cultures were pretreated with vehicle or various concentrations of GMI (0.1 and 2 μg/mL) for 30 min. before being incubated with LPS (100 ng/mL)/IFN-γ (10 U/mL) for an additional 4 h. Total cellular proteins were extracted and subjected to Western blot analysis with indicated antibodies (**A**). Nuclear proteins were extracted and subjected to EMSA for the measurement of NF-κB, AP-1, CREB, and Stat1 DNA binding activities (**B**). One representative blot of four independent culture batches is shown and the fold of relative protein content or complex content is depicted under the blots. The protein content or complex content of untreated control was defined as 1.0. The + means the groups treated with LPS/IFN-γ and the rest groups without + indicate vehicle treatment. * *p* < 0.05 vs. untreated control and # *p* < 0.05 vs. LPS/IFN-γ control.

**Figure 9 ijms-19-03678-f009:**
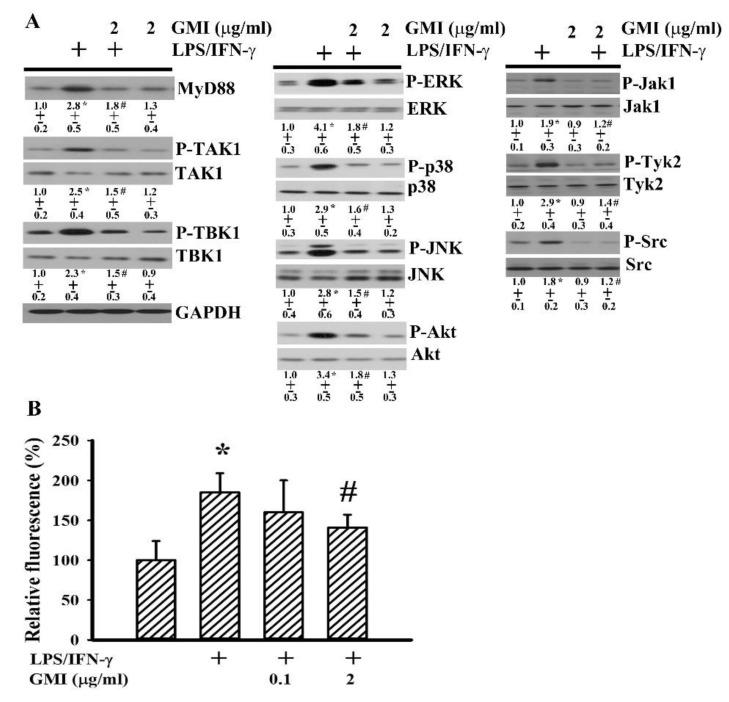
GMI alleviated intracellular signaling molecule activation. Neuron/glia cultures were pretreated with vehicle or various concentrations of GMI (0.1 and 2 μg/mL) for 30 min. before being incubated with LPS (100 ng/mL)/IFN-γ (10 U/mL) for an additional 4 h. Total cellular proteins were extracted and subjected to Western blot analysis with indicated antibodies (**A**). One representative blot of four independent culture batches is shown and the fold of relative protein content is depicted under the blots. The protein content of untreated control was defined as 1.0. The levels of intracellular free radicals were measured by the oxidation of 2′, 7′-Dichlorodihydrofluorescein Diacetate (H2DCFDA) (**B**). The fluorescent signal of untreated control was defined as 100%. * *p* < 0.05 vs. untreated control and # *p* < 0.05 vs. LPS/IFN-γ control, *n* = 4. The + means the groups treated with LPS/IFN-γ and the rest groups without + indicate vehicle treatment.

**Figure 10 ijms-19-03678-f010:**
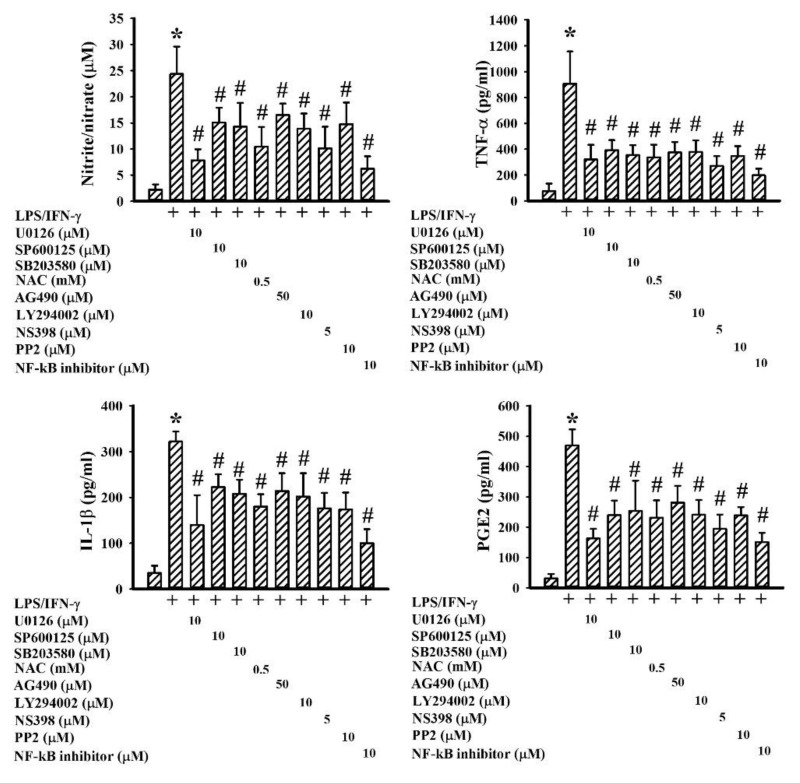
Pharmacological inhibitors alleviated inflammatory mediator production. Neuron/glia cultures were pretreated with vehicle, U0126 (10 μM), SP600125 (10 μM), SB203580 (10 μM), NAC (0.5 mM), AG490 (50 μM), LY294002 (10 μM), NS398 (5 μM), PP2 (10 μM), or NF-kB inhibitor (10 μM) for 30 min. before being incubated with LPS (100 ng/mL)/IFN-γ (10 U/mL) for an additional 24 h. Supernatants were collected and subjected to Griess reagent or ELISA for the measurement of NO, TNF-α, IL-1β, and PGE2. * *p* < 0.05 vs. untreated control and # *p* < 0.05 vs. LPS/IFN-γ control, *n* = 4. The + means the groups treated with LPS/IFN-γ and the rest groups without + indicate vehicle treatment.

**Figure 11 ijms-19-03678-f011:**
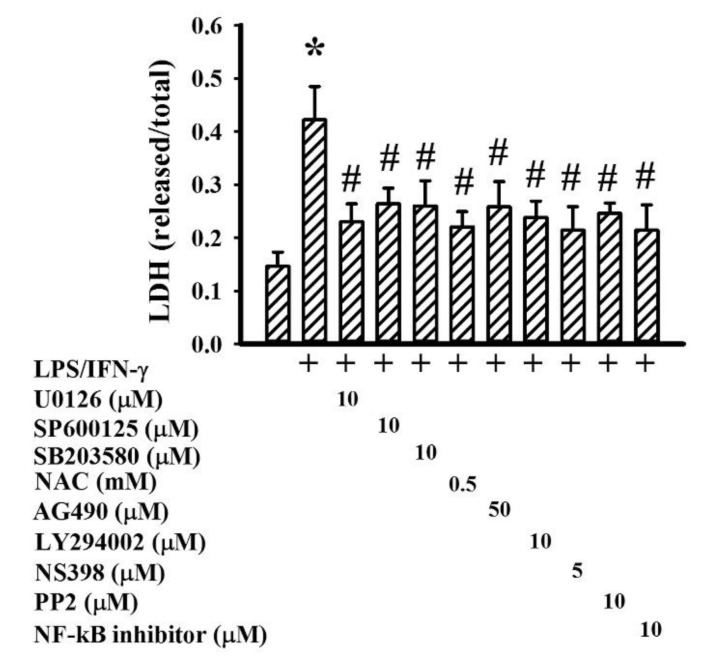
Pharmacological inhibitors alleviated neuronal cell death. Neuron/glia cultures were pretreated with vehicle, U0126 (10 μM), SP600125 (10 μM), SB203580 (10 μM), NAC (0.5 mM), AG490 (50 μM), LY294002 (10 μM), NS398 (5 μM), PP2 (10 μM), or NF-kB inhibitor (10 μM) for 30 min. before being incubated with LPS (100 ng/mL)/IFN-γ (10 U/mL) for an additional 48 h. Cell damage was measured by LDH efflux assay. * *p* < 0.05 vs. untreated control and # *p* < 0.05 vs. LPS/IFN-γ control, *n* = 4. The + means the groups treated with LPS/IFN-γ and the rest groups without + indicate vehicle treatment.

**Figure 12 ijms-19-03678-f012:**
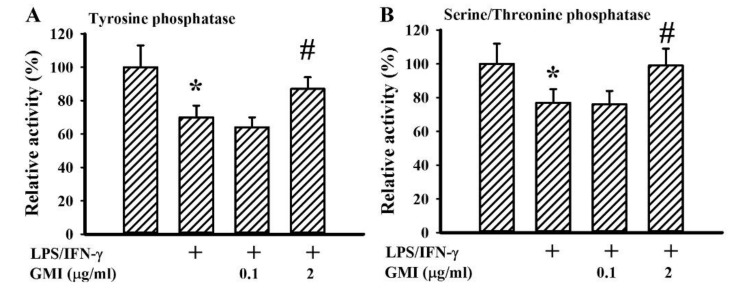
GMI alleviated protein phosphatase activity inactivation. Neuron/glia cultures were pretreated with vehicle or various concentrations of GMI (0.1 and 2 μg/mL) for 30 min before being incubated with LPS (100 ng/mL)/IFN-γ (10 U/mL) for an additional 4 h. Whole cell lysates were extracted and subjected to enzymatic assay for the measurement of tyrosine phosphatase (**A**) and serine/threonine phosphatase (**B**) activity. The activity of untreated control was defined as 100%. * *p* < 0.05 vs. untreated control and # *p* < 0.05 vs. LPS/IFN-γ control, *n* = 4. The + means the groups treated with LPS/IFN-γ and the rest groups without + indicate vehicle treatment.

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
