# Peer review of "Anti-inflammatory and Neuroprotective Effects of Fungal Immunomodulatory Protein Involving Microglial Inhibition"

_ijms, 2018, doi:10.3390/ijms19113678_

Round 1
Reviewer 1 Report
Chen et al. examined anti-inflammatory and neuroprotective effects of GMI in several culture models. The authors have concluded that GMI is potentially useful for the treatment of neuroinflammaton-mediated neurodegenerative diseases.
Comment 1: Major concern is that the effect of GMI on astrocytes were not completely excluded. The effect of GMI should be also examined in astrocyte cultures to confirm that microglia is the major target of GMI.
Comment 2: Related to comment 1, it would be better to show immunostained images of glial cells (both astrocytes and microglia) in Fig. 1 to confirm the target of GMI.
Comment 3: I think that the authors need to do some in vivo experiments to confirm the anti-inflammatory effects of GMI against LPS.
Comment 4: The density of map2-positive cells should be presented in Fig. 1.
Author Response
Comments and Suggestions for Authors
Chen et al. examined anti-inflammatory and neuroprotective effects of GMI in several culture models. The authors have concluded that GMI is potentially useful for the treatment of neuroinflammaton-mediated neurodegenerative diseases.
Comment 1: Major concern is that the effect of GMI on astrocytes were not completely excluded. The effect of GMI should be also examined in astrocyte cultures to confirm that microglia is the major target of GMI.
Responses:
In addition to neuron (MAP-2) (Figure 1A), immunocytochemical staining of astrocyte (GFAP) and microglia (CD68) was conducted and representative micrographs and quantitative data were added as Figure 2. Besides, data of cytokine production from astrocyte cultures were also added as Figure 6. Unlike microglia, astrocytes were inert to LPS/IFN-g and GMI treatment for the production of cytokines. Please refer to the revised version of manuscript.
Comment 2: Related to comment 1, it would be better to show immunostained images of glial cells (both astrocytes and microglia) in Fig. 1 to confirm the target of GMI.
Responses:
In addition to neuron (MAP-2) (Figure 1A), immunocytochemical staining of astrocyte (GFAP) and microglia (CD68) was conducted and representative micrographs and quantitative data were added as Figure 2. Besides, data of cytokine production from astrocyte cultures were also added as Figure 6. Unlike microglia, astrocytes were inert to LPS/IFN-g and GMI treatment for the production of cytokines. Please refer to the revised version of manuscript.
Comment 3: I think that the authors need to do some in vivo experiments to confirm the anti-inflammatory effects of GMI against LPS.
Responses:
We appreciated the suggestion by the reviewer. We apologized that the in vivo study was not conducted in this manuscript. However, the limitation of this manuscript and the demand of in vivo study were mentioned and discussed in the section of Discussion. Please refer to the revised version of manuscript.
Comment 4: The density of map2-positive cells should be presented in Fig. 1.
Responses:
For the quantification of immuno-positive cell number, the total number of reactive cells was counted from randomly selected four fields in a well of a 24-well plate. Four replicates were conducted for each experiment. Data of quantitative results of MAP-2-, CD68-, and GFAP-positive cell number were added under corresponding micrographs. Please refer to the revised version of manuscript.
Reviewer 2 Report
Summary:
The manuscript by Chen et al. investigated the protective effects of fungal immunomodulatory protein via inhibition of microglial polarization by several molecules. This manuscript is nicely written and includes a lot of experiments. The data presented are potentially interesting although some concerns were raised regarding the experimental data, the interpretation of results and the methods of the manuscript.
Major Compulsory Revisions
1. GMI alleviated neuronal death
LPS/IFN-γ stimulation was essential for inflammation in this studies. Why not did the authors show LPS/IFN stimulation data? In addition, neuron-glial communication via CSF-1 and IL-34. How do the author think about these factors. If the authors performed mixed culture between neuron and glial cells, evaluation of these factors was essential.
2. There were many Western blotting data. However, the authors did not show semiquantitative densitomeries’ data. Graphs were needed to confirm the results in Fig 1B, Fig 5, Fig 6 and Fig 7A.
3. The authors evaluated free radical production by the oxidation of H2DCFDA. They showed the results of fluorescent signals, although they did not show the exact figures of images.
4. The authors analyzed many comparisons by ANOVA posthoc Dunnett in Figure 8 and 9. Significant differences of some might be existed, However, Dunnett analysis is enable to compare the just control and interventions. The reviewer cannot accept the results. Statistical specialists can help the data analyses.
5. For microglial isolation, the authors performed shaking method for 24 h. Long time shaking also lift off astrocytes and microglia (Milner 2008 JCBFM). Purity study of microglia is needed.
Minor Compulsory Revisions
1. In abstract, the abbreviation GMI is needed to show full spellings for first usage.
Author Response
Comments and Suggestions for Authors
The manuscript by Chen et al. investigated the protective effects of fungal immunomodulatory protein via inhibition of microglial polarization by several molecules. This manuscript is nicely written and includes a lot of experiments. The data presented are potentially interesting although some concerns were raised regarding the experimental data, the interpretation of results and the methods of the manuscript.
Major Compulsory Revisions
1. GMI alleviated neuronal death
LPS/IFN-γ stimulation was essential for inflammation in this studies. Why not did the authors show LPS/IFN stimulation data? In addition, neuron-glial communication via CSF-1 and IL-34. How do the author think about these factors. If the authors performed mixed culture between neuron and glial cells, evaluation of these factors was essential.
Responses:
In this study, the effects of LPS/IFN-g on neuron viability, microglia viabilty, astrocyte viability, cytokine production, free radical production, phosphatase activities, and intracellular signaling molecules were measured. We appreciated the suggestion by the reviewer. The effects of CSF-1 and IL-34 might be interesting. Next study, we might investigate the change and effect of CSF-1 and IL-34. Please refer to the revised version of manuscript.
2. There were many Western blotting data. However, the authors did not show semiquantitative densitomeries’ data. Graphs were needed to confirm the results in Fig 1B, Fig 5, Fig 6 and Fig 7A.
Responses:
The quantitative data were added under corresponding blots of Western blot. Please refer to the revised version of manuscript.
3. The authors evaluated free radical production by the oxidation of H2DCFDA. They showed the results of fluorescent signals, although they did not show the exact figures of images.
Responses:
In this study, the fluorescent signals of H2DCFDA were measured by a fluorometer. We apologized that we had not obtained the fluorescent images. Please refer to the revised version of manuscript.
4. The authors analyzed many comparisons by ANOVA posthoc Dunnett in Figure 8 and 9. Significant differences of some might be existed, However, Dunnett analysis is enable to compare the just control and interventions. The reviewer cannot accept the results. Statistical specialists can help the data analyses.
Responses:
Some statistical analyses were conducted by Tukey post-hoc test. Please refer to the revised version of manuscript.
All statistical results are presented as the mean ± standard deviation. A one-way analysis of variance was performed to evaluate experimental values between groups with a consequent Dunnett’s test or Tukey post-hoc test performed for the purpose of comparison. It was considered statistically significant when the P value was less than 0.05.
5. For microglial isolation, the authors performed shaking method for 24 h. Long time shaking also lift off astrocytes and microglia (Milner 2008 JCBFM). Purity study of microglia is needed.
Responses:
The description of cell composition analysis was added. The methodology was referred to our previously reported papers. Please refer to the revised version of manuscript.
Microglia were isolated from confluent mixed glia through shaking at a speed of 200 rpm for 24 hours, and maintained in DMEM/F12 containing 10% FBS. The retained astrocytes were replated and maintained in DMEM/F12 containing 10% FBS. Neuron/glia consisted of 30-40% neurons, 40-50% astrocytes, and 10-15% microglia. Mixed glia were composed of ~85% astrocytes and ~15% microglia. The purity of microglia and astrocytes was more than 95%. Cell composition was identified and estimated by immunocytochemical staining with antibodies recognizing neuron [Microtubule-associated Protein 2 (MAP-2), Transduction Laboratories, Lexington, KY], astrocyte (Glial Fibrillary Acidic Protein, GFAP, Santa Cruz Biotechnology, Santa Cruz, CA), and microglia (CD68, Santa Cruz Biotechnology, Santa Cruz, CA), respectively.
Minor Compulsory Revisions
1. In abstract, the abbreviation GMI is needed to show full spellings for first usage.
Responses:
The writing was changed. Please refer to the revised version of manuscript.
Herein, we investigated the anti-inflammatory and neuroprotective potential of fungal immunomodulatory protein extracted from Ganoderma microsporum (GMI) in an in vitro rodent model of primary cultures.
Round 2
Reviewer 2 Report
The reviewer accepted their corrections.